# Degradation of Poliovirus Sabin 2 Genome After Electron Beam Irradiation

**DOI:** 10.3390/vaccines13080824

**Published:** 2025-07-31

**Authors:** Dmitry D. Zhdanov, Anastasia N. Shishparenok, Yury Y. Ivin, Anastasia A. Kovpak, Anastasia N. Piniaeva, Igor V. Levin, Sergei V. Budnik, Oleg A. Shilov, Roman S. Churyukin, Lubov E. Agafonova, Alina V. Berezhnova, Victoria V. Shumyantseva, Aydar A. Ishmukhametov

**Affiliations:** 1Institute of Biomedical Chemistry, 10 Pogodinskaya str., 119121 Moscow, Russia; a.shishparyonok@yandex.ru (A.N.S.); ivin_uu@chumakovs.su (Y.Y.I.); pinyaeva_an@chumakovs.su (A.N.P.); agafonovaluba@mail.ru (L.E.A.); alinalikeed@mail.ru (A.V.B.); viktoria.shumyantseva@ibmc.msk.ru (V.V.S.); 2Department of Biochemistry, People’s Friendship University of Russia Named After Patrice Lumumba (RUDN University), 6 Miklukho-Maklaya str., 117198 Moscow, Russia; 3Chumakov Federal Scientific Center for Research and Development of Immune-and-Biological Products of Russian Academy of Sciences, 8/1 Polio Institute Settlement, Moskovsky Settlement, 108819 Moscow, Russia; kovpak_aa@chumakovs.su (A.A.K.); levin_iv@chumakovs.su (I.V.L.); ishmukhametov@chumakovs.su (A.A.I.); 4Teocortex LLC, 34/6 Pervomaysky Settlement, 108808 Moscow, Russia; sbudnik@teocortex.com (S.V.B.); oshilov@teocortex.com (O.A.S.); rchuryukin@teocortex.com (R.S.C.); 5Department of Biochemistry, Pirogov Russian National Research Medical University, 1 Ostrovityanova str., 117997 Moscow, Russia

**Keywords:** poliomyelitis virus, accelerated electrons, genome degradation, biosensor, electrochemical analysis

## Abstract

Objectives: Most antiviral vaccines are created by inactivating the virus using chemical methods. The inactivation and production of viral vaccine preparations after the irradiation of viruses with accelerated electrons has a number of significant advantages. Determining the integrity of the genome of the resulting viral particles is necessary to assess the quality and degree of inactivation after irradiation. Methods: This work was performed on the Sabin 2 model polio virus. To determine the most sensitive and most radiation-resistant part, the polio virus genome was divided into 20 segments. After irradiation at temperatures of 25 °C, 2–8 °C, −20 °C, or −70 °C, the amplification intensity of these segments was measured in real time. Results: The best correlation between the amplification cycle and the irradiation dose at all temperatures was observed for segment 3D, left. Consequently, this section of the poliovirus genome is the least resistant to the action of accelerated electrons and is the most representative for determining genome integrity. The worst dependence was observed for the VP1 right section, which, therefore, cannot be used to determine genome integrity during inactivation. The electrochemical approach was also employed for a comparative assessment of viral RNA integrity before and after irradiation. An increase in the irradiation dose was accompanied by an increase in signals indicating the electrooxidation of RNA heterocyclic bases. The increase in peak current intensity of viral RNA electrochemical signals confirmed the breaking of viral RNA strands during irradiation. The shorter the RNA fragments, the greater the peak current intensities. In turn, this made the heterocyclic bases more accessible to electrooxidation on the electrode. Conclusions: These results are necessary for characterizing the integrity of the viral genome for the purpose of creating of antiviral vaccines.

## 1. Introduction

The development of inactivated polio vaccines based on Sabin strains was the most significant achievement in the 70-year history of poliomyelitis vaccines after the introduction of wild-strain-inactivated and Sabin-strain-attenuated vaccines [1]. The primary goal of attenuation strategies is to reduce virulence and maintain high immunogenicity in the resulting viral vaccine particles. The development of vaccine-associated paralytic polio as a side effect of oral Sabin-polio-attenuated vaccine limits their use and reduces their effectiveness [2]. Despite this, its widespread use in the 1960s–2000s led to a significant decrease in the incidence of paralytic poliomyelitis worldwide and the victory over the circulation of wild strains of poliovirus types 2 and 3 [3].

The World Health Organization’s Global Polio Eradication Initiative encourages vaccine manufacturers to use Sabin strains to develop and produce inactivated vaccines to minimize the risk of spreading wild strains [4].

Inactivating agents must destroy viral structural, immunogenically unimportant proteins or the viral genome to prevent the infection of cells by a virus and/or completely destroy its ability to replicate in infected cells [5]. The reversion to virulence has been documented for some polio vaccines, raising concerns about the completeness of genome degradation after virus inactivation for vaccine purposes [6]. Moreover, the presence of wild-type and vaccine-derived polio virus in sewage and the possibility of mutation to its original virulence raises question about safe, fast and affordable methods for virus inactivation in wastewater [7,8].

Formaldehyde [9] or beta-propiolactone [10] can be used for the chemical inactivation of poliovirus Sabin strains. Formaldehyde is used in the current production of inactivated polio vaccines. It is used to inactivate purified and concentrated viral suspension at the final stage of the technological process before formulation [1]. The use of formaldehyde is known to alter the antigenic epitopes of the virus [5]. Also, both chemical agents are toxic, so that the chemical agent should be removed and/or converted into a nontoxic substance before the formulation of final vaccine formulation [11]. Therefore, alternative methods of virus inactivation are being developed, and radiation technologies can provide the destruction of the viral genome (RNA in poliovirus) and maintain high immunogenicity by preserving the protein antigen (D-antigen in poliovirus) [12]. The inactivation of Sabin poliovirus strains with accelerated electrons is attractable because of low requirements for infrastructure in comparison to gamma or ultraviolet irradiation technologies [10,12]. Poliovirus genomic RNA is sensitive to radiation damage because it exists in a single copy [13], and because of the breaks in its chain stop replication. According to radiation theory, the proteins of immunogenic D-antigen (VP1, VP2, and VP3) seem to be more resistant to accelerated electrons than RNA molecules due to their high molecular mass, high number of weak van der Waals bonds and the number of copies in a viral unit [14,15].

Analysis of viral genome integrity after inactivation is routinely performed by quantitative polymerase chain reaction (qPCR) [16,17,18]. The cycle threshold (Ct) of qPCR is defined as the number of amplification cycles required for the accumulated fluorescence (resulting from the amplification of target DNA) to reach a threshold value above background. Ct values are therefore inversely proportional to the stability of the viral genome; low Ct values indicate genome integrity and high Ct values indicate a high degree of viral genome degradation [19]. The main limitation of qPCR is its ability to give reliable results only with short amplicons in the range of 150–500 base pairs [20]. The size of the genome of different viruses varies from 2000 nucleotides to 1.2 million bases [21], and in most studies there is no argument to explain the choice of site for amplification to detect genome breakdown after inactivation [16,17,18].

Many important processes in living cells (organisms) involve electrons as donors or acceptors [22]. Various strategies for the electrochemical detection of viruses and viral diseases based on genosensor and immunosensor approaches have been described in the literature [23,24,25,26]. Electroanalysis allows the registration of nucleotide molecular profiles (after isolation of DNA, RNA from biosamples) by the electrooxidation of heterocyclic bases [27,28], which makes it possible to perform comparative analysis relative to the control (intact sample) and to evaluate the mechanism of interaction of various chemical compounds with nucleotides [29,30,31], as well as to analyze DNA or RNA decomposition. Previously, we developed a label-free approach based on the differential pulse voltammetry (DPV) technique for the direct detection of DNA fragmentation and degradation after exposure to restriction enzymes and during apoptosis [22].

In this study, we have shown that different sites of poliovirus RNA have a different dependence of Ct on the dose of accelerated electron irradiation. The degradation of the viral genome was also analyzed by an electrochemical approach. These results are necessary to confirm genome decomposition in viral particles after inactivation for antiviral vaccine development purposes.

## 2. Materials and Methods

### 2.1. Poliomyelitis Virus Propagation

Poliovirus of the Sabin strain type 2 was propagated in Vero cells (WHO 10-87). Vero cells were grown in a 50 L bioreactor XDR-50 (Cytiva, Marlborough, MT, USA) using the LXMC-dex1 microcarrier 3 g/L (SunResin, Shaanxi, China) in the MEM (Institute of Poliomyelitis, Moscow, Russia), containing 5% fetal serum (LTBiotech, Vilnius, Lithuania). Cultivation settings were as follows: temperature—37 °C, pH control—7.3 ± 0.1; DO (dissolved oxygen) control—70%, stirring speed—40 rpm. Before infection, the suspension of microcarriers with cells was washed with Hanks’ solution (Institute of Poliomyelitis, Moscow, Russia) and the nutrient medium was changed to M199 (Institute of Poliomyelitis, Moscow, Russia). Cells were infected with a multiplicity of infection of 0.01–0.05 TCID/cell after reaching a cell concentration of 1–2 × 10^6^ cells/mL. Infected cells were incubated at 34 °C for 2 days (or until the monolayer of cells on the microcarriers degrades). Viral suspension was clarified by filtration through the PES-filter cascade 0.65/0.45–0.22 µm and concentrated with a Sartoflow Advanced (Sartorius, Göttingen, Germany) tangential flow filtration system with 100 kDa PES cassettes. During inactivation, the viral suspension did not contain infected cells or their fragments. Virus titer was measured via a 50% tissue culture infectious dose assay [32] using Hep-2 cells. For the experiments, a virus suspension with a concentration of 10^10^ TCID_50_ was used.

### 2.2. Inactivation by Electron Beam Irradiation or Chemicals

Inactivation of Sabin 2 polioviruses was performed by accelerated electron beam irradiation at doses 5, 10, 15, or 25 kGy. The power accelerator was 15 kW and the energy of electrons was 10 mEV. Samples with viral material were packed by 4 mL into 5 mL cryotubes (three tubes for each inactivation condition) and placed in hermetically sealed containers. The samples were initially precooled and irradiated at temperatures 25 °C, 2–8 °C, −20 °C or −70 °C. Control samples were not irradiated and were stored at the same temperatures. Actual absorbed doses were determined by absorbed dose detector SO PD (F) R-5/50 (VNIIFTRI, Solnechnogorsk, Russia). Three independent irradiations were performed for each dose and temperature.

To inactivate viruses with formaldehyde, the pool of virus-containing fractions after chromatographic purification was diluted in a 2:1 ratio with 199 medium, a 37% formalin solution was added to a final concentration of 0.025% (*v*/*v*), and this was mixed and filtered through a Millipore membrane (Merch-Millipore, Kenilworth, NJ, USA) with a pore diameter of 0.22 microns (for sterilization and removal of viral aggregates). Inactivation was performed at a temperature of 37 ± 1 °C for 13 days. The neutralization of formaldehyde was carried out using sodium sulfite (Na_2_SO_3_).

>To inactivate viruses by beta-propiolactone, the suspension containing the poliovirus was inactivated by beta-propiolactone with a final concentration of 0.2% (*v*/*v*). After the addition of beta-propiolactone, the contents were mixed and transferred to fresh sterile containers to remove the virus that had not reacted with beta-propiolactone. The mixture was kept for 24 h at 4 °C and then incubated at 37 °C for three hours. Beta-propiolactone was neutralized by the same method as for formaldehyde [33]. Three independent inactivations by chemicals were performed.

### 2.3. Determination of Residual Infectious Activity

To test the completeness of inactivation of viral samples after irradiation with accelerated electrons, we used the 2-step passaging method in Vero cell culture. The test sample was diluted 5 times with DMEM (Institute of Poliomyelitis, Moscow, Russia) to 10 mL and added to a 25 cm^2^ flask containing a monolayer culture of Vero cells. The cells were incubated with the samples for 5 days at 34 °C. The cell monolayer was inspected using a light microscope (ICX41, Sunny Instruments, equipped with ADF LIVE 4K camera, 100× magnification). Each cell monolayer after incubation with the sample at each passage was compared with a negative control (cells not exposed to samples or virus) and a viral control—cells in a medium supplemented with viable virus at a concentration of 10 TCID_50_/mL. If there were signs of specific degradation of the cell monolayer, as in the viral control sample, the sample was considered to contain infectious poliovirus. If there were no signs of degradation, the culture medium was diluted 5 times in DMEM to 10 mL. The resulting sample was incubated in a 25 cm^2^ flask with a monolayer of Vero cells for 5 days (second passage). Intact Vero cell monolayer at the second passage indicates the absence of infectious virus in the original test sample.

### 2.4. Determination of D-Antigen

The D-antigen content of the poliovirus was analyzed using an Enzyme-Linked Immunosorbent Assay (ELISA) based on purified polyclonal rabbit antibodies specific for D-antigen. Affinity-purified IgG was obtained by immunizing rabbits with the purified poliovirus antigen. IgG was adsorbed in a 96-well panel (Corning Costar, Corning, NY, USA) for 20 h at a temperature of 2–8 °C. After the adsorption procedure, the wells were washed twice with a washing solution (0.01 M phosphate-buffered saline, pH 7.4, with 0.05% Tween-20), and blocked using a blocking solution (0.01 M phosphate-buffered saline, pH 7.4) with 1% fetal bovine serum (FBS) for 1 h at a temperature of 37 °C. Samples were diluted in ELISA buffer (0.01 M phosphate-buffered saline, pH 7.4, with 0.05% Tween-20 and 1% FBS) and added to the wells of the plate and incubated for 2 h at 37 °C. The plate was washed twice with the washing solution. Poliovirus-specific rabbit IgG, conjugated with biotin, was used for D-antigen detection by incubation with samples for 1 h at 37 °C. After washing of the plate with the washing solution, streptavidin peroxidase conjugate (Sigma, St. Louis, MO, USA) was added to the wells and incubated for 1 h at 37 °C. Tetramethylbenzidine solution (TMB solution, Sigma, St. Louis, MO, USA) was used for detection, and was incubated for 15 min. The results were recorded using an iMark microplate spectrophotometer (Bio-Rad, Hercules, CA, USA). International NIBSC standard (National Institute for Biological Standards and Control, Herts, UK) was used as a standard sample of D-antigen content. The D-antigen content in non-radiated control samples was considered to be 100%.

### 2.5. Quantitative Polymerase Chain Reaction (qPCR)

Viral RNA was isolated from inactivated or control samples using the Quick-RNA Viral Kit (Zymo Research, Seattle, WA, USA) according to the manufacturer’s protocol. The concentration of isolated RNA was determined using the Qubit RNA HS Assay Kit (Thermo Fisher Scientific Inc., Waltham, MA, USA) with the Qubit 4 Fluorometer (Thermo Fisher Scientific Inc., Waltham, MA, USA). The integrity and quality of RNA was determined using the Qubit RNA IQ Assay Kit (Thermo Fisher Scientific Inc., Waltham, MA, USA). Isolated RNA was visualized by electrophoresis in 1% agarose gel, stained with ethidium bromide, and photographed under UV light. Five micrograms of viral RNA was subjected to reverse transcription using MMLV RT kit (Evrogen, Moscow, Russia) in 25 µL reaction mixture. The hexamer random primer was used for the initiation of transcription. The viral genome was segmented to 20 regions to obtain amplification segments (see Appendix A in the Appendix A). The location of PCR segments on the viral genome is shown in Figure 1. Primers listed in Table 1 were used for qPCR of each segment.

Primer sequences were selected based on the following principles. (1) Protein-centric principle, meaning that the chosen pair of primers must cover the region encoding the single functional protein and not that located on the border site of two proteins; (2) the length of the amplifying segment is no longer than 500 bp (as this is a limitation of PCR technique); (3) if the genome region encoding a single functional protein is longer than 500 bp, it is divided it into several (2 or 3) amplicons; and (4) the optimal quality of primers is as follows: GC%—50–60, self-complementarity not higher than 4, self 3′ complementarity not higher than 2, melting temperature difference not higher than 3%, annealing temperature is 58–60 °C, and length 20–22 nucleotides.

Real-time PCR detection was performed in 20 µL mix-HS SYBR (Evrogen, Moscow, Russia) using the DTprime5 machine and software (DNA Technology, Protvino, Russia). The Ct of detection was defined for each sample as the number of amplification cycles required for the accumulated fluorescence to reach a threshold value above background. The Ct of each sample was measured in triplicate.

### 2.6. Electrochemical Study

Electrochemical measurements were performed at room temperature in 60 μL of 0.1 M potassium phosphate-buffered saline (PBS) containing 50 mM NaCl (pH 7.4) using a PalmSens potentiostat (PalmSens BV, Houten, The Netherlands) with PSTrace software (version 5.8). The following DPV parameters for direct electrochemical oxidation of RNA were used: potential range 0.2–1.2 V, pulse amplitude 0.025 V, potential step 0.005 V, pulse duration 50 ms, and scan rate 0.05 V/s. The screen-printed electrodes (SPE, ColorElectronics, Moscow, Russia) with a graphite working electrode were modified by 2 μL (0.75 ± 0.05 mg/mL) water dispersion of 0.4% single-wall carbon nanotubes (SWCNT TUBALL BATT H_2_O, OCSiAl, Novosibirsk, Russia). The SPE/SWCNT electrodes were kept at room temperature until completely dry. All other chemicals were of analytical grade and purchased from Sigma-Aldrich (St. Louis, MO, USA). Modified electrodes were pre-treated three times by DPV in PBS, pH 7.4. Two microlitres of the RNA probe was dropped onto the surface of the modified electrode and incubated for 24 h at +4 °C before measurements. All potentials were referenced to the Ag/AgCl reference electrode. PSTrace with baseline correction was used to process the signal intensity values. All electrochemical data presented are the mean of three experiments with standard deviation.

### 2.7. Statistics

The graph of Ct versus dose was plotted and the coefficient of determination (R^2^) was calculated using Graph Prism 8 software (GraphPad Software, Inc., Boston, MA, USA). Results are presented as mean ± standard deviation. R^2^ value higher than 0.7 was considered significant.

## 3. Results

### 3.1. Residual Infectious Activity and D-Antigen Content of Irradiated Samples

We monitored residual infectious activity in irradiated samples using a two-step passaging method in Vero cell culture. Samples irradiated at doses 10–25 kGy at 25 °C were completely inactivated (Table 2, Figure 2). However, samples treated at 2–8 °C or −20 °C were only inactivated by doses of 25 kGy. None of the doses were capable of inactivating samples frozen at −70 °C, as all of them showed residual infectious activity.

The content of D-antigen was determined in completely inactivated samples. No preserved D-antigen was found in inactivated samples irradiated at 25°C; however, it was detected in samples irradiated at 2–8 °C or −20 °C.

Samples inactivated by formaldehyde or by beta-propiolactone did not show residual infectious. The recovery of their D-antigen was 40.0 ± 10.0% for formaldehyde-treated samples and was not detected in beta-propiolactone-treated samples.

The results of this experiment demonstrated that a temperature of 25 °C is not optimal for the inactivation of poliovirus by irradiation with accelerated electron, as its D-antigen was not preserved. Decreasing the temperature to 2–8 °C or −20 °C resulted in the recovery of the D-antigen, and a lower temperature was associated with higher antigen recovery.

### 3.2. Quality of RNA Isolated from Irradiated Virus

We studied the integrity and quality of the isolated viral RNA by fluorescently labeling the large and small RNAs. A number between 1 and 10 represents the percentage of large RNA molecules in the sample. We observed a gradual decrease in IQ values in samples irradiated with accelerated electron doses (Figure 3A). The IQ of the non-irradiated control samples was 7.4–8.8 AU, indicating a high degree of RNA integrity. This value dramatically decreased to 0–0.8 AU in 25 kGy irradiated samples, indicating complete RNA degradation. In general, the IQ values of samples irradiated at −20 °C or −70 °C were higher than those of samples irradiated at 25 °C or 2–8 °C, indicating that the viral genome is more stable in frozen samples when exposed to electron beam irradiation. The IQ values of samples inactivated by formaldehyde or beta-propiolactone did not exceed 4 AU (Figure 3B). Agarose gel electrophoresis of total RNA isolated from irradiated viral samples (Figure 3C) confirmed that the genome was degraded by irradiation. The main RNA bend from the control, non-irradiated sample was located at the top of the gel, within the 7 kb molecular weight marker area. After irradiation with 10 kGy, the size of the main bend decreased to 5 kb. Subsequent irradiation with increased doses of radiation resulted in the complete loss of full-size RNA. The RNA bends from samples inactivated with formaldehyde or beta-propiolactone are located near the 3 kb molecular weight marker (Figure 3D).

### 3.3. 3D Left Segment of Poliovirus Most Sensitive to Accelerated Electrons

Values of Ct are inversely proportional to the stability of the viral genome [19]. Using qPCR, we determined Ct values for each viral genome segment in samples irradiated at different doses and temperatures. R^2^ values for the dose-dependency of Ct showed that decreasing sample temperature was associated with the increased stability or integrity of viral RNA. The highest R^2^ values were observed for samples irradiated at 25 °C or 2–8 °C (Figure 4A). The mean R^2^ values among all samples were 0.693 and 0.706, respectively. The lowest R^2^ values were for samples frozen at −20 °C (mean 0.243) and −70 °C (mean 0.153).

Low Ct values indicate genome integrity and high Ct values indicate a high degree of viral genome decomposition. Only the 3D left segment of the viral RNA reached an R^2^ value higher than 0.7 at all temperatures (Figure 4A,B). The highest R^2^ values of 0.912 and 0.864 were observed for this segment in samples irradiated at 25 °C or 2–8 °C. Viral segment VP 1 right (Figure 4A,C) showed the lowest R^2^ values; even for a temperature of 25 °C it had an insignificant value of 0.251.

The results of this experiment show that the 3D left segment of the viral genome is the most representative for studying the effect of accelerated electrons on genome degradation, while the VP1 left segment is the least representative.

We determined Ct for each poliovirus genome segment after inactivation with the commonly used chemical reagents formaldehyde or beta-propiolactone (Figure 5). Very good consistency between Ct values was observed for samples treated with each of these reagents. Again, the left 3D segment showed the highest Ct values of 32.10 ± 0.10 for formaldehyde and 32.03 ± 0.15 for beta-propiolactone-treated samples, demonstrating the high sensitivity of this segment to inactivating chemicals. The VP1 right segment showed the lowest Ct values of 20.87 ± 0.06 for formaldehyde and 20.87 ± 0.15 for beta-propiolactone-treated samples. Low Ct values again indicate the highest resistance of this segment to degradation.

### 3.4. Electroanalysis of Poliovirus RNA Degradation on SPE/SWCNT

The electrochemical approach for the analysis of intact viral RNA and irradiation-inactivated RNA samples is based on a comparative analysis of their nucleotide profiles, which makes it possible to assess the effect of irradiation parameters and draw conclusions on the degree of RNA fragmentation. The electrochemical profiling of RNA isolated from samples of the Sabin 2 strain of poliovirus before and after inactivation by accelerated electrons was carried out. Nucleotide molecular fingerprints were obtained by differential pulse voltammetry. Two oxidation peaks at potentials of +0.49 ± 0.01 V (with shoulder at +0.60 ± 0.01 V) and +0.82 ± 0.01 V corresponding to electrooxidation of RNA nucleobases guanine and adenine, respectively, were registered (see Appendix A in the Appendix A). The electrooxidation peak at potential +0.49 ± 0.01 V is not uniform and composite. It is possible to assume that the wave with a broad shoulder not only corresponds to guanine, but also represents the oxidation of proteins that may form a complex with virus RNA (ribonucleoproteins or covalently bound protein VPg) [34] or the RNA fragments of different length with different guanine exposure and, respectively, different guanine availability for electrochemical oxidation. For a quantitative comparison of the electrooxidation peak current (I) at a potential of +0.49 ± 0.01 V, which revealed the most intense response, this experimental parameter was normalized by the RNA concentration (C) in the samples, I/C (Figure 6). As shown in Figure 6A–D, the increase in the irradiation dose was accompanied by an increase in the signals of RNA electrooxidation (R^2^ from 0.49 to 0.93). The absence of a linear dependence of the signals of RNA electrooxidation on the irradiation dose at 25 °C and freezing temperatures (−20 °C and −70 °C) is probably due to the suboptimal irradiation conditions of the samples. In addition, irradiation at room temperature is usually accompanied by uncontrolled radiolysis of water. The highest R^2^ values were observed for samples irradiated at 2–8 °C (R^2^ = 0.93, Figure 6A). The peak current of guanine electrooxidation (Figure 6A) increases eightfold when the RNA is irradiated (irradiation dose 25 kGy, 2–8 °C) and compared to the control native RNA. We assume that an increase in peak current intensity is a result of strand viral RNA fragmentation or complete disruption during the irradiation procedure [35,36].

The data obtained suggest that the increase in RNA electrooxidation signals with increasing irradiation dose is associated with damage to the RNA structure. The results obtained are consistent with the qPCR results for these samples and the results of gel electrophoresis.

## 4. Discussion

Currently, most viral vaccines are produced using chemical inactivation. Among these vaccines, we can highlight those against tick-borne encephalitis [37], hepatitis A [38] and poliomyelitis [1]. All of them were produced using formaldehyde inactivation. Meanwhile, vaccines against COVID-19 were produced using beta propiolactone inactivation [39]. From a technological standpoint, the use of chemical inactivators does not require complex and expensive equipment; only mixers that maintain a certain temperature are needed. On the other hand, the use of physical inactivation methods requires complex equipment: emitters of electrons or other particles, and devices for ultraviolet irradiation. This complicates the process chain and increases the cost of production. Methods of virus inactivation using gamma irradiation or irradiation with accelerated electrons have been described, with dozens of patents registered [40]. However, these technologies have not been implemented in production, likely due to the challenges associated with the equipping of pharmaceutical enterprises with irradiators. Perhaps, as technology advances and irradiators become more accessible, safe and user-friendly, this inactivation will be adopted in the industry.

Accelerated electron irradiation has an advantage in terms of the energy of free electrons. We utilized an accelerator that enabled us to reach an energy of 10 megaelectron volts. In comparison, the energy of UV irradiation electrons is only around 10 eV at a wavelength of 254 nm [41,42]. High electron energy values allow for faster virus inactivation (the same dose is accumulated in less time) and also enable the inactivation process to be carried out through a thicker layer of liquid containing the virus. UV virus inactivation on an industrial scale requires special devices that require the creation of a stable thin liquid layer for photons to pass through. The appearance of air bubbles or other impurities adversely impacts the efficiency of UV inactivation. Accelerated electron irradiation does not require special conditions to prevent the formation of viral conglomerates or air bubbles.

According to modern classification, the poliovirus is called Enterovirus coxsackiepol and belongs to the Picornaviridae family, Ensavirinae subfamily, Enterovirus genus. D-antigen particles of poliovirus consist of four capsid proteins: VP1, VP2, VP3, and VP4. Among these, VP4 is situated on the inner surface of the capsid. VP2 and VP4 are generated through the cleavage of the precursor protein VP0. The surface of the capsid is formed by 60 protomers of the proteins VP1, VP2 and VP3. The viral genome is a non-segmented single-stranded RNA molecule of positive polarity, 7.4 kb in length, with a viral protein associated with the genome Vpg (virus protein genome linked) at the 5′ end and a poly(A) sequence at the 3′ end [43]. It should be noted that the packaging of poliovirus genomic RNA is replication-mediated and there are currently no known packaging signals within regions of the poliovirus genome that would indicate their interactions with capsid proteins [44].

Genome decomposition ensures the inactivation of viruses during the development and production of vaccines based on attenuated viruses. Fine-tuning the type of irradiation, dose, and regimen, such as energy and temperature, can result in the selective decomposition of viral nucleic acids. According to the radiation target theory, viral genomes are more susceptible to structural damage by electron beam irradiation than viral proteins due to their higher molecular weight [45]. Based on these data, viral RNA was chosen as the experimental target for the analysis of polio virus Sabin 2 genome degradation.

The results of this study demonstrated three approaches that can be applied to analyze viral genome decomposition after the irradiation of inactivation by chemicals. This study demonstrated three approaches for analyzing viral genome decomposition after chemical inactivation or irradiation. The first approach is based on measuring RNA fluorescence in treated samples using a commercially available kit that detects the proportion of intact and decomposed RNA in a sample (Figure 3A,B). As expected, the integrity (i.e., IQ value) decreased upon irradiation with accelerated electrons. These results were consistent with the electrophoretic visualization of degraded RNA, which showed a decrease in the size of RNA bends after irradiation (Figure 3C,D). It should be noted that IQ values also depended on sample temperature: lower temperatures corresponded to higher IQ in samples irradiated with the same doses. These results could be explained by the ability of accelerated electrons to disrupt nucleic acids through direct impact, which induces strain breaks, or indirectly by inducing reactive oxygen species during water radiolysis [46]. Radiolysis decreases at lower temperatures, which leads to a decrease in genome decomposition rates. In our experiments, the dose of accelerated electrons had a greater effect on genome decomposition than temperature did.

The second approach demonstrated in this study is the possibility of determining RNA integrity through qPCR of its different segments. PCR is a frequently used technique for demonstrating genome disintegration upon virus inactivation [16,17,18]. However, as our work shows, the choice of the genome segment to be amplified is of primary importance for obtaining valid results (Figure 4 and Figure 5). We have demonstrated that certain regions of the genome are more stable for irradiation, while others are highly susceptible to degradation. As yet we still do not have a reasonable explanation for why different Sabin 2 genome segments have different resistance to degradation. Some structural features of RNA folding within viral particles may be significant in exposing the 3D left segment and hiding the VP1 right segment from accelerated electrons and/or products of water radiolysis.

The relative resistance of certain fragments of the genome (for example, VP1 right) to irradiation with accelerated electrons, as well as chemical methods of inactivation, can theoretically be explained by the possible interaction of these regions with capsid proteins inside the particle. A closely located protein shell can “shield” and non-specifically protect a region of the genome from chemical or physical effects. However, to date, specific fragments of the poliovirus genome that participate in binding to capsid proteins and are responsible for the packaging of genomic RNA have not been identified [44]. Moreover, the packaging process is apparently primarily associated with the replication process and is possibly triggered by protein–protein interactions (for example, between capsid proteins and the replication protein 2CATPase) [47]. But this fact does not negate the fact that individual fragments of the genome have a higher affinity for the inner part of the capsid, while others have a lower affinity. Perhaps our data are an indirect indication of the involvement of more stable fragments in the genome–capsid interaction.

The quality of the primers used may also have an effect on the efficiency of PCR. Currently, we do not have a reasonable explanation for why some segments have shown rather high Ct values in non-irradiated samples. High Ct values in control samples decreased R^2^ values.

Traditional laboratory-based nucleic acid assays possess many shortcomings and are time-consuming, labor-intensive and need additional expensive chemical and biochemical reagents and modern complicated equipment [34]. Electrochemical DNA/RNA biosensors have drawn attention due to their obvious advantages, such as high sensitivity, portability, cost effectiveness, fast response time, and compatibility with miniaturized detection technologies and compact equipment with user-friendly software [48,49]. The DPV method using carbon nanotube-modified electrodes was developed to study the effect of electron beam irradiation potency on the damage of viral genome RNA. The third approach used in our study involved the comparative electrochemical profiling of RNA isolated from intact and irradiated viruses. The electrochemical profiles have reflected the extent of RNA damage depending on the dose of accelerated electron irradiation and the temperature regime of the experiments (Figure 6). The increase in RNA peak current intensity due to electrooxidation signals with increasing irradiation dose is associated with damage to viral RNA structure, due to the greater availability of heterocyclic bases to electrochemical reactions at the electrode [35,36]. The differential pulse voltammetry approach confirmed the quantitative PCR (qPCR) results for the degradation of the polio virus Sabin genome after electron beam irradiation at an experimental temperature of 2–8 °C and an irradiation dose of 20–25 kGy. In our experimental assay, we demonstrated that electrochemical label-free biosensors proved the concept of viral genome degradation, as registered by the IQ assay and qPCR.

## 5. Conclusions

In conclusion, this study demonstrated that irradiation of the Sabin 2 poliovirus by accelerated electrons leads to the degradation of viral genome segments at different rates. Some segments demonstrated rather high R^2^ values and can be considered representative for degradation study, while R^2^ values for other segments were rather low and are not representative for degradation research. This indicates that the accelerated electron impact process on RNA is uneven, and the most representative segment for the qPCR study must be selected for each virus before studying virus inactivation.

## Figures and Tables

**Figure 1 vaccines-13-00824-f001:**
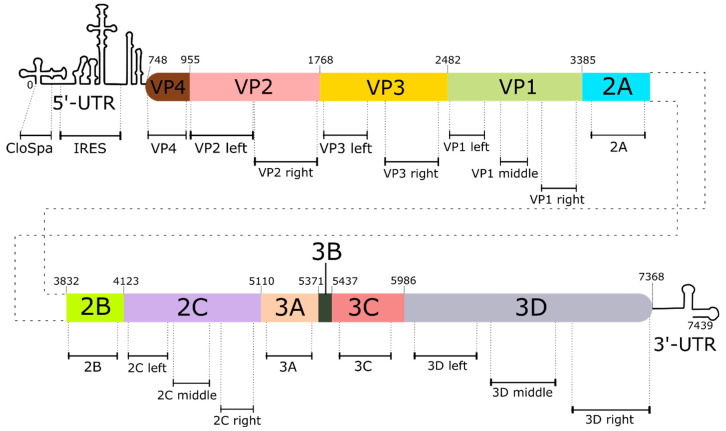
Diagram of the location of PCR segments on the genome of the Sabin strain type 2 of poliovirus. The diagram shows the position of the 5′-end of each of the regions encoding viral proteins. UTR is the non-translating region.

**Figure 2 vaccines-13-00824-f002:**
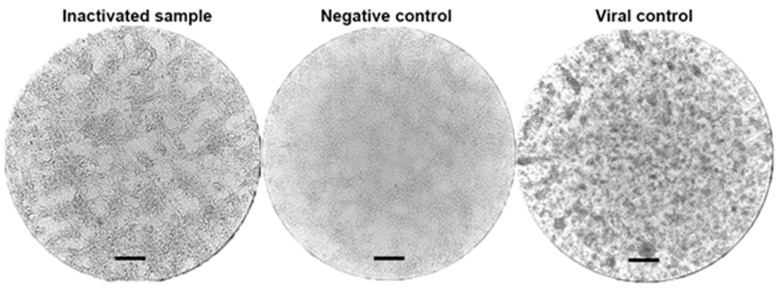
Representative photos of light microscopy show a monolayer of Vero cells during the control of residual infectivity. Inactivated sample, example of Vero cell monolayer after incubation with an inactivated sample of poliovirus strain Sabin type 2 (irradiated with accelerated electrons at a temperature of 2–8 °C and a dose of 25 kGy). Negative control, an intact monolayer of Vero cells that did not undergo treatment with the studied samples or virus. Viral control, the state of the monolayer of Vero cells after infection with poliovirus strain Sabin type 2 with 10 TCID_50_/mL infectious titer. Scale bars in the images correspond to 250 µm.

**Figure 3 vaccines-13-00824-f003:**
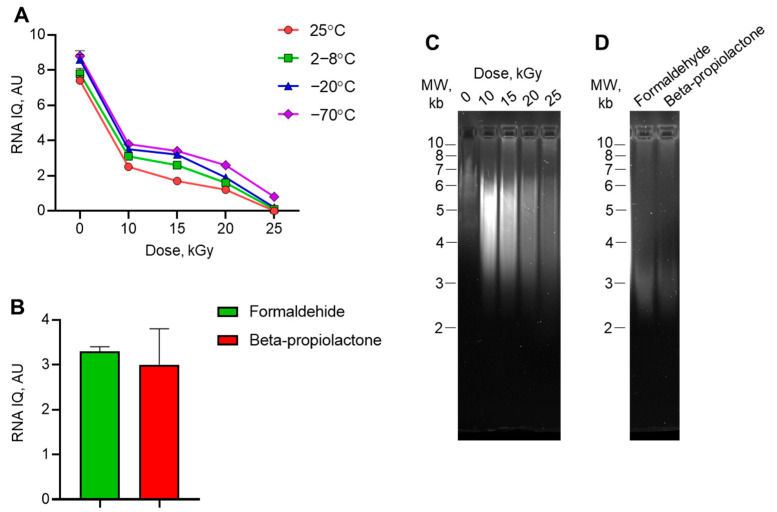
RNA quality assessment in treated poliovirus. Total RNA was isolated from (**A**) irradiated samples or (**B**) samples inactivated by formaldehyde or beta-propiolactone. RNA integrity and quality (IQ) was measured by fluorescence and scored in arbitrary units (AU). N = 3. Agarose gel electrophoresis showing the degradation of viral RNA (**C**) after irradiation and (**D**) after chemical inactivation.

**Figure 4 vaccines-13-00824-f004:**
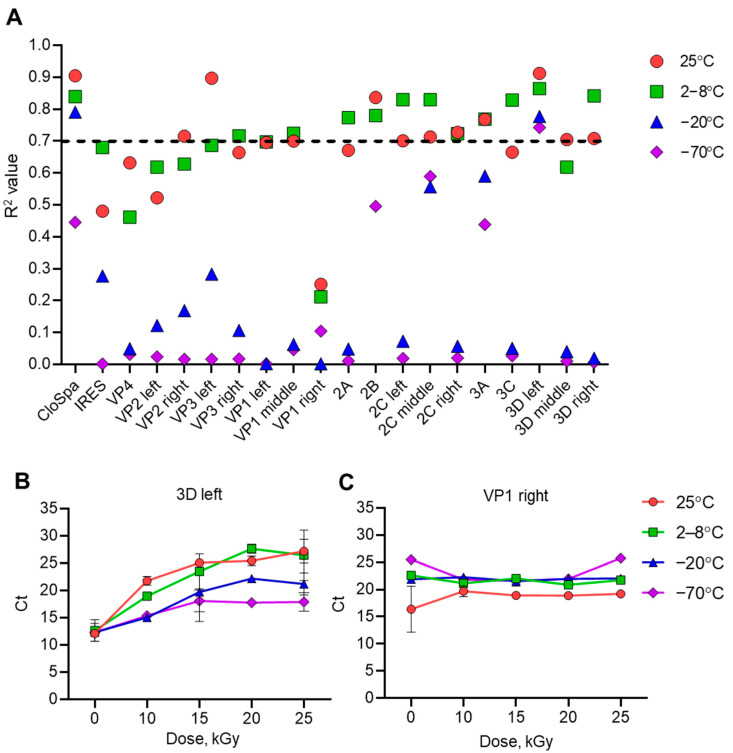
The dependence of the cycle threshold (Ct) on the irradiation dose. (**A**) The R^2^ value showing the sensitivity of poliovirus RNA irradiated at different temperatures to accelerated electrons. The Sabin 2 strain samples were irradiated with accelerated electrons within the range of 10–25 kGy (50 mEV). Total RNA was isolated and subjected to qPCR. Plots of Ct versus irradiation dose were made and R^2^ values were calculated. Black horizontal dashed line shows R^2^ = 0.7. Plots of Ct versus irradiation dose for (**B**) the most representative 3D left and (**C**) the least representative VP1 right sites of viral RNA. The plots for the other viral RNA segments are presented in Appendix A in the Appendix A.

**Figure 5 vaccines-13-00824-f005:**
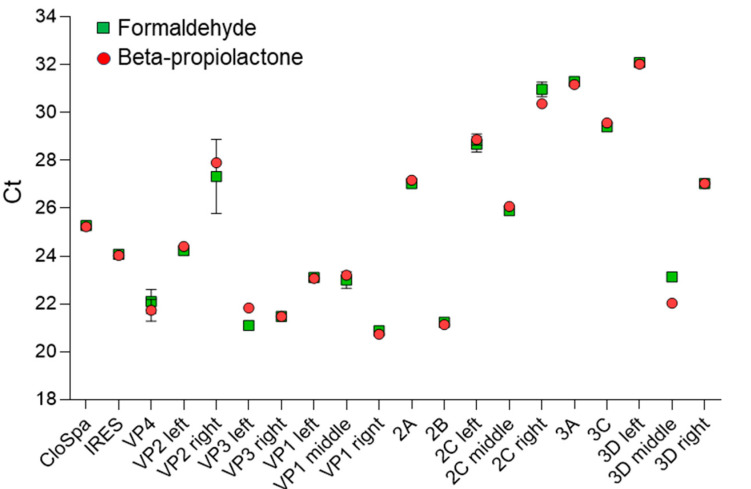
Cycles threshold (Ct) for Poliovirus genome segments after inactivation with formaldehyde or beta-propiolactone.

**Figure 6 vaccines-13-00824-f006:**
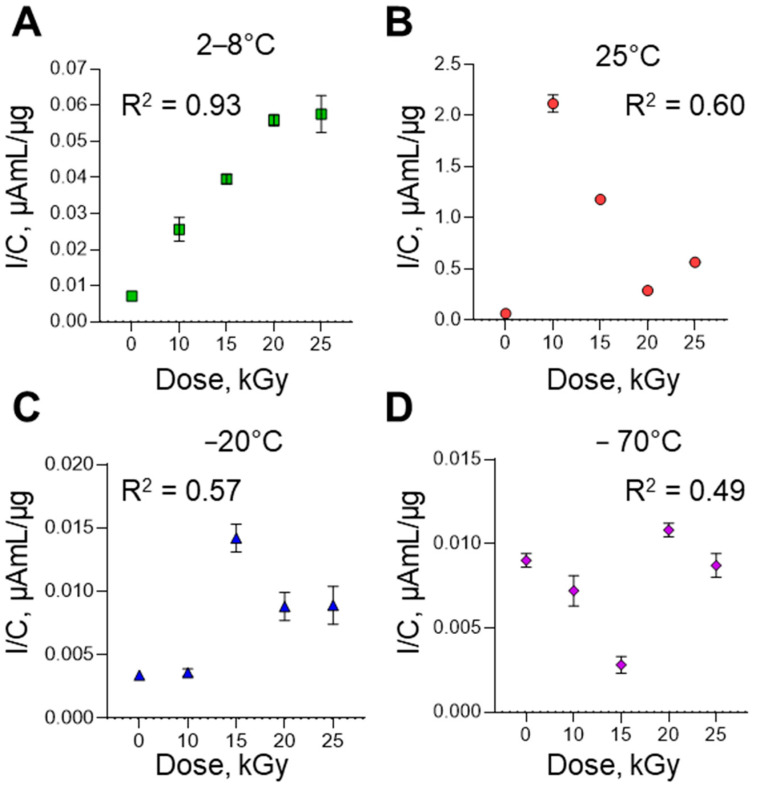
The dependence of the I/C signals for poliovirus Sabin 2 RNA on the irradiation doses at different temperatures to accelerated electrons: (**A**) 2–8 °C; (**B**) 25 °C; (**C**) −20 °C; (**D**) −70 °C. Plot of the I/C poliovirus irradiation doses with the best R^2^ = 0.93 for the temperature to accelerated electrons of 2–8 °C. The zero point reflects the DPV signal of intact (control) poliovirus RNA without irradiation procedure. I—peak current, µA, C—RNA concentration, µg/mL.

**Table 1 vaccines-13-00824-t001:** List of primers used for qPCR.

#	Region Name	Sense Primer (5′-3′)	Antisense Primer (5′-3′)	Product Size, bp	Start Position
1	CloSpa	aacagctctggggttgtacc	tggtttcgtgcttctaagttgc	118	5
2	IRES	tccccggtgacattgcatag	caaagtagtcggttccgcca	373	176
3	VP4	gcgcccaagtttcatcacag	tttagcatgggagcggtctta	202	752
4	VP2 left	tccccaaacattgaggcgtg	tctggaactgcaaacacccc	386	955
5	VP2 right	agaatgcgaatccaggcgaa	tggcacagtgatgttgcgta	371	1388
6	VP3 left	gtaaccagtacctgaccgca	gagagacacaagatcggcgt	229	1796
7	VP3 right	ttttgcggctcaatgatggc	tatctcgcagtaagcgcaca	325	2125
8	VP1 left	tggtgacatgattgaggggg	gactctgatcgcgttcgtct	219	2486
9	VP1 middle	cggacatgcattgaaccaagt	attcccacgtagggcactga	162	2922
10	VP1 right	gtaccactagcgggtcaagc	cccttttctggtagtggggt	261	3142
11	2A	gcaaaatgccgtgagtgtca	ctgctccatagcttcctcctc	367	3465
12	2B	tgagtcacttggtgctgcat	ttctttagccactgccacgg	231	3849
13	2C left	ctgcaaagggactggagtgg	agtggtgcaaacctcttgga	237	4163
14	2C middle	gagcaagcaccgtattgagc	tgcctttctcttctagcgagg	287	4467
15	2C right	accaactccagtcggatcac	aactgaattgccttgccaca	212	4786
16	3A	ccctccggagtgtatcaacg	gcccagcgaacagcttgta	209	5157
17	3C	aggccctgggtttgattacg	ggtttgtcgtccaccgagat	397	5436
18	3D left	aaccaaacttgaacccagcg	tccgcatttccttggtgtct	362	6051
19	3D middle	gtggagcagggaaaatccag	ggcatgccgccttttacg	359	6487
20	3D right	tacccccatgaggttgatgc	acggcggtacaatgtagagt	369	6985

**Table 2 vaccines-13-00824-t002:** D-antigen recovery in inactivated (green color) and control non-radiated samples (red color).

Dose, kGy	Temperature, °C			
	25	2–8	−20	−70
0	100%	100%	100%	100%
10	0	ND	ND	ND
15	0	ND	ND	ND
20	0	ND	ND	ND
25	0	27.5 ± 3.0	46.3 ± 5.2%	ND

ND, no data.

## Data Availability

The data presented in this study are contained within the article.

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
