# Peer review of "Degradation of Poliovirus Sabin 2 Genome After Electron Beam Irradiation"

_vaccines, 2025, doi:10.3390/vaccines13080824_

Round 1
Reviewer 1 Report
Comments and Suggestions for Authors
The manuscript discusses the effects on the integrity of the viral genome after irradiation with accelerated electrons. Using the Sabin 2 model polio virus, researchers segmented the viral genome to identify its most sensitive and radiation-resistant parts. Segment 3D left showed the best correlation between amplification cycles and irradiation doses, indicating it is the most sensitive part to electrons and most suitable for assessing genome integrity. Conversely, segment VP1 right was the most resistant part and therefore unsuitable for this purpose. Additionally, an electrochemical method was used to compare viral RNA integrity before and after irradiation, revealing that increased irradiation doses led to more significant RNA strand breakage. Overall, these findings are important for understanding viral genome integrity in the context of different inactivation methods and could contribute to the characterization of inactivated antiviral vaccines. THowever, the impact of accelerated electrons on the viral RNA is irregular, and the best suited segment for qPCR analysis after inactivation must be selected for each virus before.
The manuscript is clearly structured and, with few exceptions, comprehensibly written. However, some minor points should be addressed by the authors.
- the authors should describe the method for electroanalysis used in addition to qPCR to characterize viral RNA after irradiation in a more comprehensible way. Since in Figure 5 a linear correlation is observed only for the temperature range 2-8°C and not for the other temperatures, this might be confusing to the reader. It appears that at 25°C and -20°C the damage to RNA is higher at low doses than at high doses. Although the authors say that this is due to the lower radiolysis at lower temperatures, this casts doubt on the application of the method for the low temperature ranges.
- It is not clearly described how many independent irradiations were performed to generate the data. Since the damage to the RNA tends to occur in an untargeted manner, several independent irradiations would be absolutely essential to be able to make a valid statement about the sensitivity of the investigated genome fragments.
- The dose-dependent influence of electron radiation on various gene segments is well described by the authors. It is pointed out that this is a marker for inactivated virus vaccines. However, it is not shown whether the maximum applied dose of 25 kGy actually leads to complete inactivation. For gamma irradiation, significantly higher values (45 kGy in Tobin et al., 2020 https://doi.org/10.1371/journal.pone.0228006 ) were used. In order to be able to really apply this in the context of vaccine production, this proof should be provided. Alternatively, at least the titer reduction for the doses investigated should be shown. This would also make it possible to determine a D10 value in order to determine the necessary dose for complete inactivation. It would be interesting to see how the identified gene fragments then behave at this dose if it is higher than 25 kGy.
- Irradiation with electrons is currently not a method which is used for the production of licensed vaccines, although it has great potential. The inactivation method has so far only been described in the literature. In order to underpin the relevance of the present study for vaccine production, this should be made clear in the discussion.
- The following should also be changed in the text:
Material and Methods:
Line 139: Typo at Quick
Line 149: Cyrillic font at unit description
Line 151: Full stop at end of sentence missing
Line 236: „Very good agreement between Ct values was observed for samples treated with each of these reagents”
-->The term “correlation” or ‘consistency’ would be better here than “agreement”.
Line 254: "The electrooxidation peak at a potential of +0.49±0.01 V is not uniform and composite. It can be assumed that the wave with the foreign shoulder does not correspond only to guanine....."
-->Please insert “to” in front of guanine.
Line 266: Please correct the typo in the word disruption.
Line 290: “....detects the ratio of integrated and decomposed RNA in a sample.”
--> Please change to intact instead of integrated
Line 347: "Some segments (primary 3D left segment) are rather sensitive while some are resistant 347 (primary VP1 right segment)."
--> Sentence is double in the text. Please delete one sentence.
Reviewer 2 Report
Comments and Suggestions for Authors
Overall
The authors present an interesting study on the degradation of the Sabin 2 poliovirus genome after electron beam irradiation, assessing the genome integrity using multiple molecular and electrochemical approaches. The focus on alternative inactivation methods is valuable given the limitation of current inactivation methods in conventional IPV.
However, I have several key concerns and recommendations that should be addressed to make the manuscript scientifically robust. Moreover, since inactivation method is directly related to translational relevance, cross-evaluation of the inactivation process is crucial.
Major Comments
- A critical aspect of poliovirus inactivation for vaccine is not only the genome degradation but also the preservation of D-antigenicity (as described by the authors in lines 64-67), which is directly related to immunogenicity and protective efficacy. The authors state that immunogenic D-antigen proteins are more resistant to accelerated electrons, but I cannot find detailed supporting information in the cited references (ref #11, #12).
1-1. Please provide the appropriate references to support this statement.
1-2. The results lack any direct measurement of D-antigen content or its preservation post-irradiation. This information is crucial to determine whether this method is suitable for vaccine use.
- Electrochemical Assay is an elegant method in principle and the authors have thoroughly addressed results and discussion. However, the authors should include (or at least propose) systematic cross-validation with conventional infectivity assay before supporting its practical application for vaccines. For example, complete inactivation must be demonstrated by residual infectivity assay, such as multiple blind passages of the electron beam-exposed virus on susceptible cell lines. Especially since electron impact on RNA is uneven and this approach is newly proposed for poliovirus, this validation process should be added to strengthen authors’ conclusions.
- While the manuscript briefly mentions other genome fragmentation or inactivation methods, it does not provide an comparison between electron beam and other methods for poliovirus, particularly other irradiation approaches, in the Discussion section. The author should discuss these comparative advantages and disadvantages in detail.
Additionally, the results show differences between chemical- and electron beam- inactivated viruses. Please describe in detail the significance and implications of these findings in the discussion section.
Minor Comments
- It is interesting to see variability in segment degradation (e.g. 3D left and VP1 right), but it is not fully mechanistically explained. Given the figure 3 and the statement in the Discussion (lines 309-311), have the authors predicted RNA structures with any available tool(s)?
- Several section numberings in the results section in incomplete. Please correct them.
- R square is useful value to see coefficient. But if some groups show R2 value above 0.7, how can we interpret the statistical significance of these differences? If it is possible, please provide additional statistical information on the graphs or results.
- How did the authors choose the primer sets for qPCR? What rationale was used to select these specific genome regions?
- Please check the references in the overall manuscript. If the cited reference(s) cannot support the contents of the manuscript, please revise them accordingly.
Reviewer 3 Report
Comments and Suggestions for Authors
The purpose of this study of virus irradiation with accelerated electrons was to determine the degree of damage to the viral genome as a measure of the degree of inactivation of the virus. Using Sabin poliovirus 2, the integrity of the genome was assessed over a range of dosages using virus at four temperatures and compared to effects of inactivation by formalin or beta-propiolactone. Using the Qubit RNA IQ assay, the integrity of the genome declined from an untreated value of 7-9 to 3-4 with a dosage of 10kGy and near 0 with 25 kGy while the standard treatment with formaldehyde or beta-propiolactone left the IQ at approximately 3. This suggests considerable degradation of the 7 kb genome for either method. Assessment of genomic degradation by differential pulse voltammetry suggests that only at when the virus irradiation is done at 2-8oC is there a consistent relationship between degradation and irradiation dosage.
The genomic integrity was assessed using 20 different quantitative RT-PCR assays specific for 20 sites in the viral genome and the degree of loss of genomic integrity measured as Ct at each site per accelerated electron dosage was determined. Without treatment or irradiation, Ct values for each assay of the viral RNA mostly varied from 15 to 20. The highest Ct value (using the 2-8oC samples), indicating the lowest concentration of amplified cDNA, was approximately 23 for the VP1 right amplicon (nt 3142-3402). The lowest Ct value prior to treatment, indicating the highest concentration was 13 from the 3D left amplicon (nt 6051-6412). Examination of degradation across the genome with this method for formaldehyde or beta-propiolactone generated produced a wide variance of Ct from the amplified segments from 21 to 32.
R2 was determined for each of these plots and used as an assessment of the dependence of Ct upon radiation dose. R2 for these Cts versus irradiation dosage demonstrated highest R2 from irradiations done at 25oC and 2-8oC and had means around 0.7. At all temperatures used, 3D left had the highest R2 and VP1 right had the lowest. 3D left thus had the most representative degradation versus irradiation dosage plot and the plot of VP1 right demonstrated that Ct did not vary significantly according to the irradiation dosage. These two amplicons were the extremes of Ct also with the formaldehyde or beta-propiolactone treated virus.
The main finding of this study was that the RNA oxidation (as measured with differential pulse voltammetry) with increasing irradiation does reflect viral RNA genome damage as demonstrated by the relationship between increasing Ct and increasing dosage. The authors also discuss the low R2 of the VP1 right amplicon versus irradiation and suggest that this derives from resistance to irradiation of this segment within the viral capsid. The authors have not determined the efficiency of the different quantitative PCR of cDNA derived from the irradiated or chemically treated virus. It seems given that the VP1 right amplicon has the lowest amount of amplification (highest Ct value) without any treatment that this reaction is very low in efficiency. With a low efficiency it is more likely that the assay would not vary much with treatment of either kind. While the authors have found the assay 3D left is very sensitive and likely to provide an adequate assay of genomic integrity (given that any break that is within the IRES or open reading frame will prevent infectivity), this might have been demonstrated by determining the efficiency of each of the segment assays. This efficiency needs to be demonstrated with a control cDNA for the genome for each of the sets of primers. Then the quality and sensitivity of each assay can be compared.
Reviewer 4 Report
Comments and Suggestions for Authors
This is a reasonably good and very interesting paper as it touches on a novel idea of using electrons to further attenuate the Sabin poliovirus strain. I have however some questions that need to be addressed or have to be explored.
1) The issue of current Sabin vaccine is not just the potential of virulence to some individuals. The attenuated virus can revert to its original virulent strain. Therefore, the more mutations made to the Sabin vaccine the safer it becomes.
https://www.nature.com/articles/s41541-023-00740-9
2) What advantages do electron blasting of the virus offer over exposure of the virus to light especially UV-light? Lights are photons and they have similar quantum mechanical properties as electrons. In fact, UV is one of the ways of attenuating viruses. The authors may want to do further literature search for this to answer my question:
https://pmc.ncbi.nlm.nih.gov/articles/PMC8879476/
3) Related to (1), it not just about finding better techniques to attenuate the virus. There is a problem of Sabin strain accumulation in the sewage system and the possibility of mutation to its original virulent self. Electron blasting may offer a cheaper alternative to chemical or X-ray radiation. The problem with electron blasting of wastewater is that the virus may be embedded in fecal matter, which protects the virus form electron exposure:
https://pmc.ncbi.nlm.nih.gov/articles/PMC8310065/
https://pmc.ncbi.nlm.nih.gov/articles/PMC10917565/
4) From what I understand after reading the Methods section, the virus was exposed to electron blasting in the absence of cells upon filtration. This should be emphasised. My question as related to (3) is: Did the authors exposed the virus-infected cells to electrons and did it have any results? This is important as in (3) because it reflects on the viability of electron exposure as a mean for virus elimination in wastewater.
5) Attempts to develop attenuated vaccines by randomly mutating the RNA may be more difficult than it seems because the damaged virus may no longer have the ability to replicate due to the damage. sustained. Have the authors attempted to grow virus using some of the damaged viruses? It would be interesting to see the results.
6) There is no description of the poliovirus. RNA/DNA virus? Family? Brief description of the virion physiology or the viral proteins. Description of the proteins is important as many of the proteins are protecting the RNA from damage. How the respective viral protein(s) fold to encapsulate the RNA also matters on what segment of the RNA is easily damaged.
7) There is no description of how Sabin vaccines are prepared.
Round 2
Reviewer 2 Report
Comments and Suggestions for Authors
Thank you for the authors’ responses and the revisions made to the manuscript. Many concerns I raised have been resolved in the revised manuscript. However, there are still a few points where it is difficult to evaluate the improvement since some information is missing.
- The data corresponding to Result 3.1 is missing in the current manuscript. Please provide the relevant figure or table and clearly indicate them within the main text.
- Please elaborate on the specific procedure used to detect infectious activity. For example, if the data for result 3.1 include monolayer cell microscope images, describe which tool was used. Additionally, to validate the assay system, it is strongly recommended to include a non-irradiated virus control as a positive control to demonstrate that the assay can detect infectious virus effectively.
- Regarding the response to minor comment 4, it would greatly improve the manuscript’s transparency and reproducibility to include the primer design criteria in the Method section. Please briefly describe the key metrics.
Reviewer 3 Report
Comments and Suggestions for Authors
I do not disagree with the response of the authors to comments 1, 2 and 3.
The response to comment 4: “The only suitable control cDNA for this type of study is a cDNA from control non-treated virus. In our study this control is labeled as 0 kGy. We measured the concentration of isolated RNA, and all samples for qPCR had an equal amount of RNA (it is a standard practice). Therefore, changes in Ct values reflect the initial amount of non-broken (intact) copies of given genome segment. When establishing R2 we are not comparing relative amplification among different genome segments. R2 instead establishes the change in Ct values of a single segment after irradiation for which 0 kGy control cDNA already has its own amplification efficiency.”
I understand that the control for this study was the Ct value from control non-treated virus (0 kGy) which was the same for all the assays. I agree that the Ct values with different kGy doses reflect the change in copy number of unbroken genomes. I examined the figure 3B and C graphs as well as those in Figure 1S. The comment I made was the correspondence of greatest Ct value of the 0 kGy using the VP1 right assay to the least R2 of the Ct versus kGy dosage. The low R2 of the VP1 right assay is demonstrative of the lack of change in the Ct with different dosages (Figure 3C). The 3D left assay of 0 kGy virus has the lowest Ct (0 kGy) of all the assays and has the highest R2 which is evident in Figure 3B in which the Ct value varies with kGy dose indicating that in this assay, the amount of damaged RNA increased with increased irradiation dose, resulting in higher Ct with higher kGy dosage and a high R2. The authors state that R2 (of the Ct versus kGy dose) establishes the change in Ct values of a single segment after irradiation. The low R2 of VP1 right corresponds to the curves in Figure 3C in which the Ct does not vary appreciably with the kGy dose. The 3D left assay of 0 kGy virus has the lowest Ct of all the assays and has the highest R2 which is evident in Figure 3B in which the Ct value varies with kGy dose indicating that in this assay, the amount of damaged RNA increased with increased irradiation dose, resulting in higher Ct with higher kGy. The Ct of 3D left (control 0 kGy) is 13 for irradiation at 2-8oC and that for VP1 right is 22.5 indicating a very different amount of amplification of cDNA. As both samples had equal amounts of viral genome, the assay comparing Ct and kGy dosage is much more likely to have an accurate correspondence of dosage and amplification of viral genome with the 3D left assay than with the VP1 right assay. To say that the genome location is more important than the ability of the primer pairs to amplify cDNA would require a comparison of two reactions which had a similar 0 kGy Ct indicating a similar degree of amplification of cDNA per viral genome.
My comment on this was that a low efficiency assay as VP1 right must be (given the high Ct of the 0 kGy control) may be less able to provide an accurate estimate of the amount of damaged RNA in the sample after irradiation. I think this is more likely to be the reason that the VP1 right assay has such a low R2, rather than resistance to irradiation of the viral RNA in the capsid in this region. I also think that the 3D left assay is the most efficient assay and thus has a very high R2. The authors have developed a method to detect the most efficient qPCR assay to detect the degree of viral genome damage which is the 3D left assay in the case of Sabin 2. However, the authors attribute this in the case of electron beam irradiation to sensitivity (3D left assay) or resistance (VP1 right assay) of regions of the genome within the virus. If the 0 kGy Ct of VP1 right and 3D left were similar, the authors could hypothesize that there was a difference in the extent of damage to the two regions, but this would require changing to the assay for VP1 right to a more efficient primer pair for that region. This study has demonstrated the value of the qPCR assay, but also may point to a need to determine the most efficient assay for the utility of this method, as the authors have done for Sabin 2.
The authors can discuss this point in the discussion in lines 419-431. If they cannot address the issue of the difference in the 3D left and VP1 right 0 kGy Ct, I suggest they change the sentence in the conclusion (lines 455-456): “Some segments, such as 3D left segment, are rather sensitive, while others, such as VP1 right segment, are resistant.”
Reviewer 4 Report
Comments and Suggestions for Authors
Improvements seen.
Author Response
Thank you very much for thorough consideration of our manuscript.
Round 3
Reviewer 2 Report
Comments and Suggestions for Authors
Thank you for providing additional information and revision. All previous concerns have been addressed.